# Optimization of Ultrasonic Cellulase-Assisted Extraction and Antioxidant Activity of Natural Polyphenols from Passion Fruit

**DOI:** 10.3390/molecules26092494

**Published:** 2021-04-24

**Authors:** Wei Wang, Yu-Ting Gao, Ji-Wen Wei, Yin-Feng Chen, Qing-Lei Liu, Hui-Min Liu

**Affiliations:** 1School of Perfume and Aroma Technology, Shanghai Institute of Technology, Shanghai 201418, China; wangweittg@sit.edu.cn (W.W.); gyt_18271651021@163.com (Y.-T.G.); CYF19970806@163.com (Y.-F.C.); liuqinglei2006888@163.com (Q.-L.L.); 2Engineering Research Center of Perfume & Aroma and Cosmetics, Ministry of Education, Shanghai 201418, China; weijiwen@jahwa.com.cn

**Keywords:** passion fruit, polyphenol, ultrasonic cellulase extraction, Box-Behnken Design (BBD), antioxidant activity

## Abstract

In this paper, ultrasonic cellulase extraction (UCE) was applied to extract polyphenols from passion fruit. The extraction conditions for total phenol content (TPC) and antioxidant activity were optimized using response surface methodology (RSM) coupled with a Box-Behnken design (BBD). The results showed that the liquid-to-solid ratio (X_2_) was the most significant single factor and had a positive effect on all responses. The ANOVA analysis indicated quadratic models fitted well as TPC with R^2^ = 0.903, DPPH scavenging activity with R^2^ = 0.979, and ABTS scavenging activity with R^2^ = 0.981. The optimal extraction parameters of passion fruit were as follows: pH value of 5 at 30 °C for extraction temperature, 50:1 (*w*/*v*) liquid-to-solid ratio with extraction time for 47 min, the experimental values were found matched with those predicted. Infrared spectroscopy suggested that the extract contained the structure of polyphenols. Furthermore, three main polyphenols were identified and quantified by HPLC. The results showed the content of phenolic compounds and antioxidant activity of the optimized UCE were 1.5~2 times higher than that determined by the single extraction method and the Soxhlet extraction method, which indicates UCE is a competitive and effective extraction technique for natural passion fruit polyphenols.

## 1. Introduction

Passion fruit (*P. edulis* Sims) is a native fruit plant of Brazil [1]. It has been extensively used in food, pharmaceutical, and cosmetic industries [2,3] due to its rich bioactive constituent (phenolic compounds, carotenoids, vitamins, and fibers) [4,5]. Nowadays, polyphenol compounds of passion fruit have been extensively studied as it exhibits strong antioxidant activity and good for human nutrition [6]. In addition, passion fruit extract has also been found to be useful in the cosmetics field because of its in vitro tyrosinase inhibitory effect and a certain SPF value [7]. With more and more advantages of this fruit are found, extracting bioactive compounds from passion fruit is of high interest. However, conventional extraction methods such as solvent extraction suffer from low extraction yields, energy-consuming, and low efficiency [8,9] extracting bioactive compounds from plants in an efficient way is a challenging task.

Enzyme extraction technology is becoming an alternative method to conventional technology for natural extraction because of being an efficient and eco-friendly extraction technology [10,11]. The basic principle of enzymatic extraction is to destroy the cell wall using an enzyme as a catalyst so that the effective components can be released more quickly [12]. Compare to conventional extraction technology, it can increase the extraction yield and lower the energy consumption in a more mild way [13]. Besides, it has been reported that enzyme activity can be enhanced by ultrasonic treatment under optimized conditions [14], which results in the molecular structure of cellulase enzyme changing and affecting the kinetics and thermodynamics [15]. Ultrasonic enzyme-assisted extraction has been developed as a new method for the extraction of bioactive molecules [16]. This method can not only reduce the processing time but also can enhance the quality of extracts. However, there are few studies about extracting polyphenols of passion fruit using ultrasonic cellulase extraction, especially research on how to extract polyphenols efficiently. The response surface method is an efficient experimental statistical approach commonly used for the optimization of processes and conditions [17,18]. In this experiment, Box-Behnken design is selected for experiment design because it has higher accuracy than other design models like central composite design and D-optimal design. It’s useful for fitting second-order response-surface models and sometimes runs fewer runs than central composite design [19].

Hence, the focus of this study aims to: (1) investigate the effect of ultrasonic cellulase assisted extraction on total polyphenols content and antioxidant activity, (2) optimize the extraction conditions by using response surface methodology with Box-Behnken design, (3) characterize the structure of the extract by IR and compare the common polyphenols from the passion fruit in different extraction methods by HPLC.

## 2. Results and Discussion

### 2.1. Effects of UCE Conditions on the TPC of Passion Fruit

The effect of liquid-to-solid ratio on the TPC of passion fruit was estimated at five ranging from 10 to 50 mL/g, other extraction conditions remained constant as follows: ultrasound power of 200 W; pH value of 4; cellulase amount of 4%; the temperature of 40 °C; ultrasound time of 30 min. As can be seen in Figure 1a, the TPC was raised from 9.07 ± 0.02 to 16.04 ± 1.30 mg GAE/g DW, with the increase of liquid-to-solid ratio from 10 to 50 mL/g. However, the TPC was almost unchanging when the liquid-to-solid ratio was greater than 40 mL/g. The reason was that the contact area of solvent and solid increases with the ratio of liquid-to-solid increasing, when the contact area reaches a certain value, the extraction speed would be saturated [20].

Five different cellulase amounts (2%, 4%, 6%, 8%, 10%) were employed to investigate the effect of cellulase amount on the TPC of passion fruit, and other conditions were fixed as follows: ultrasound power of 200 W; pH of 4; liquid-to-solid ratio of 40 mL/g; the temperature of 40 °C; ultrasound time of 30 min. Figure 1b showed that the TPC increased as the cellulose amount increased from 2% to 6%, but the TPC began to decline and remain unchanged when the amount of added cellulose over 6%. Cellulase could break down cellulose into smaller molecules. For cellulase reaction, extraction amount would be faster with cellulase concentration increasing. Therefore, the best cellulase amount was 6%.

Figure 1c showed the pH effect on the TPC of passion fruit. Other conditions were kept as follows: ultrasound power of 200 W; cellulase amount of 6%; liquid-to-solid ratio of 40 mL/g; the temperature of 40 °C for 30 min. The TPC increased as the initial stage, then the TPC decreased when pH exceeds 5. The possible explanation is that pH could influence cellulase conformation and substrate dissociation [21]. The optimal pH could maximize cellulase activity and cell-wall breaking. But the condition of cellulase reaction was changed that cellulase activity decreased, as further increasing of pH value. Hence, the optimal pH was 5.

The temperature will affect the TPC through acoustic cavitation and thermal effect [22]. Hence, different temperatures (30–70 °C) were chosen to optimize extraction efficiency with other conditions as follows: ultrasound power of 200 W; cellulase amount of 6%; liquid-to-solid ratio of 40 mL/g; the value of pH at 5 for 30 min. Figure 1d showed the TPC quickly increased as the increasing of temperature before 60 °C, while the TPC was not noticeable to change after at 60 °C. The augment in TPC may due to the increase in collision frequency between the substrate and cellulase enzyme at a higher temperature, however, the temperature over 60 °C, could bring vapor pressure which results in the decrease of cavitation intensity, thus impact the TPC [23].

As presented in Figure 1e, the TPC changed in ultrasound power (200–500 W), with other factors as follows: cellulase amount of 6%; liquid-to-solid ratio of 40 mL/g; pH of 5; ultrasound time of 30 min; the temperature of 40 °C. 300 W was the optimum ultrasound power to extract polyphenols, while the TPC decreased after peaking at 300 W. The enzyme could be in an active conformation because of the rupture in weak linkages like Van der Waals’ interaction or hydrogen bonds under ultrasound [24]. However, when ultrasound power exceeded, the cellulase activity decreased, it suggested that higher ultrasound intensity could damage the polypeptide chains and inactivate the enzyme [25].

To the effect of ultrasound time on the TPC of passion fruit, the extraction conditions were as follows: the liquid-to-solid ratio of 40 mL/g with the temperature of 40 °C, ultrasound power of 300 W, cellulase amount of 6%, pH value at 5; and ultrasound time was 10–50 min, respectively. According to Figure 1f, the TPC increased rapidly as the increasing of ultrasound time at a range of 10–40 min. However, the increase of the TPC was unclear any more after 40 min. The results suggested that the TPC increased before 40 min and then tended to be stable.

### 2.2. Building Models and Analyzing Statistics

The responses of the TPC and antioxidant activities for passion fruit were optimized based on the Box-Behnken design, and optimization of the UCE process was obtained by applying a second-order polynomial equation. Independent variables were temperature (X_1_), liquid-to-solid ratio (X_2_), ultrasound time (X_3_), pH value (X_4_). Response values (Y_i_) were the total polyphenols content, DPPH radical scavenging ability, and ABTS radical scavenging ability, respectively. Table 1 shows the experimental design on RSM. The results of the response surface analysis are presented in Table 2. 

As can be seen from Table 1, TPC, DPPH, ABTS extracts varied from 9.28 to 19.51 mg GAE/g DW, 105.62 to 420.14 mg VCE /100 g DW, 142.26 to 450.09 mg VCE /100 g DW, respectively. Optimization of the UCE process was obtained by applying a second-order polynomial equation. The model presents a high significance and a good fit with the experimental data of TPC and having less variation around the mean (R^2^ value = 0.928). The antioxidant activities of DPPH and ABTS showed the model had a positive effect, and the quality of fit to the second-order polynomial equation investigated with the coefficient of determination (R^2^), which was 0.979, 0.981, respectively. The second-order polynomial Equations (1)–(3) were established to demonstrate the relationship between variables and response variables:Y_TPC_ = 15.21 + 0.80X_1_ + 1.77X_2_ + 2.60X_3_ − 0.086X_4_ − 1.06X_12_ − 1.45X_13_ − 0.22X_14_ − 0.34X_23_ + 0.50X_24_ + 1.28X_34_ + 1.54X_1_^2^ − 0.53X_2_^2^−0.24X_3_^2^ − 0.47X_4_^2^,(1)
Y_DPPH_ = 279.03 − 8.14X_1_ + 121.25X_2_ + 2.16X_3_ − 15.13X_4_ − 9.21X_12_ + 0.34X_13_ − 22.60X_14_ + 11.53X_23_ + 15.46X_24_ − 20.83X_34_ − 14.99X_1_^2^ + 7.26X_2_^2^ − 0.44X_3_^2^ − 67.07X_4_^2^,(2)
Y_ABTS_ = 306.84 − 26.77X_1_ + 130.19X_2_ − 12.82X_3_ + 21.33X_4_ − 0.69X_12_ − 19.97X_13_ − 45.70X_14_ + 9.50X_23_ + 18.39X_24_ − 49.37X_34_ − 21.44X_1_^2^ + 4.04X_2_^2^ − 8.02X_3_^2^ − 23.68X_4_^2^,(3)

### 2.3. Interaction of Process Variables

The regression coefficient and variance analysis of the model are shown in Table 2. The *p*-value is used to distinguish the statistical significance of each coefficient and model. The *p*-value less than 0.05 implies model terms are significant. The P values of the three models are all less than 0.001, indicating the model proved highly significant effects. For TPC, the results demonstrated that TPC was mainly influenced by X_2_, X_3_ (*p* < 0.001), followed by X_1_^2^, X_13_ (*p* < 0.01) and X_1_, X_12_, X_34_ (*p* < 0.05). The lack of fit (F = 3.4) implies non-significant to the pure error fitting of the model, which shows the equation is better simulated and can analyze data well. 

The interactive effects of variables are presented by 3D response surface plots, Figure 2 shows three factors that had a great influence on TPC. The interaction between temperature and the liquid-to-solid ratio (X_12_) displayed significant (*p* < 0.05). The TPC increased with increasing temperature and liquid-to-solid ratio, while at a higher temperature (50–70 °C) and liquid-to-solid ratio (35:1–50:1), the interaction effect was not obvious. The reason may be that the appropriate temperature and liquid-to-solid ratio will improve the solubility and increase the mass transfer rate, but the excessive temperature will destroy the polyphenol structure, and too high liquid-to-solid ratio will affect the mass transfer efficiency. This interaction is consistent with [26]. The highly marked interaction effect (*p* < 0.001) between temperature and ultrasound time (X_13_) was observed in Figure 2b, it shows excessive temperature and long extraction time would decrease the TPC. Long extraction time in high temperature would evaporate the solvent and therefore influence the mass transfer rate, which decreases the value of TPC. Figure 2f shows the interaction between pH and ultrasound time (X_34_), which is also a significant factor due to *p* < 0.05. Other interactions like X_14_, X_23_, and X_24_ showed a negative effect in this model.

In vitro antioxidant activity was estimated via DPPH and ABTS radical scavenging assays. The experimental data for RSM in DPPH and ABTS were varied from 105.62 to 420.14 mg VCE/100 g DW, 142.26 to 450.09 mg VCE/100 g DW, respectively. Figure 3 and Figure 4 showed the interaction of factors on DPPH and ABTS by 3D response surface plots. The results demonstrated that DPPH was mainly influenced by X_2_, X_42_ (*p* < 0.001), followed by X_4_, X_14_, X_34_ (*p* < 0.001). While ABTS was largely influenced by X_1_, X_2_, X_14_, X_34_ (*p* < 0.001), followed by X_4_, X_42_ (*p* < 0.01), and X_3_, X_12_ (*p* < 0.05). The lack of fit value was 3.07 and 0.46 respectively (not significant), which means the models can be fitted well. As is shown in Figure 3c, the value of DPPH reached maximum as the temperature was 50 °C and the value of pH was 5, but over this point, DPPH radical scavenging assay began to decrease. This may due to high temperature and pH change the antioxidant bio-compounds structure, thus influence the antioxidant ability [27,28,29]. Similarly, the interaction caused by temperature and time (X_14_) was also shown in ABTS (*p* < 0.05), but was not as significant as DPPH (*p* < 0.001). Furthermore, the interaction between ultrasound time and pH (X_34_) showed a more significant positive effect in DPPH than in ABTS. In addition, the interaction between ultrasound time and pH (X_34_) was significant for all three response values, and it could be seen that TPC is correlated with antioxidant activity. Similar results were also reported by [30], which presented high antioxidant activities in other plant extract and positive correlations between the antioxidant activity and the TPC. It should be noticed that the TPC can be affected by different factors. Some researchers measured the antioxidant capacity of passion fruit extracts with the Trolox equivalent antioxidant capacity (TEAC) using DPPH and ABTS free radical scavenging methods. The results showed different fractions of passion fruit and extraction solvents have an impact on the antioxidant properties, among which DPPH was varied from 10 to 80 TACE, and ABTS was varied from 30 to 706 TACE [31].

### 2.4. Optimization and Verification of UCE 

The optimal extraction parameters to obtain maximum TPC, DPPH, and ABTS by RSM are as follows: pH value of 5 at 30 °C for extraction temperature, 50:1 (*w*/*v*) liquid-to-solid ratio with extraction time for 47 min, the predicted value is 20.96 mg GAE/g DW for TPC, 421.27 mg VCE /100 g DW for DPPH and 454.02 mg VCE /100 g DW for ABTS. Three parallel experiments were done to verify the accuracy of the optimal extraction conditions, the experimental values were 21.03 ± 0.10 mg GAE/g DW, 420.52 ± 0.06 mg VCE /100 g DW, and 453.87 ± 0.07 mg VCE /100 g DW, respectively. The experimental results have good reproducibility and are consistent with the predicted values, indicating the model could predict responses well. Therefore, this model can be used to optimize the UCE conditions to obtain TPC and antioxidant ability.

### 2.5. Comparison of UCE with Other Methods

To better estimate the advantages of the ultrasonic cellulase technique, the performance of UCE was compared to three conventional methods including cellulase extraction (CE), Soxhlet extraction (SE), and ultrasound extraction (UE). The ultrasound power was set at 300 W, liquid-to-solid ratio was 50 mL/g. The results in Table 3 showed that the TPC using UCE was significantly higher than others. The obtained polyphenols content was almost 1.5, and 2 times higher than UE and CE, respectively. It can be assumed that such a difference in extraction content can be attributed to the cavitation effect and cellulase activity induced by ultrasound. In addition, the SE required a long-time (3 h) and higher temperature (95 °C) to produce the same yield with CE. These results clearly demonstrate that UCE is a competitive and effective extraction technique for natural passion fruit properties, especially polyphenols.

### 2.6. HPLC Analysis

HPLC analysis is performed by comparing the retention time of the standards and the extracts. The composition of selected standards for HPLC analysis is based on the literature review of common polyphenol compounds that have been reported in passion fruit. The HPLC chromatograms of passion fruit extracts obtained by four methods are shown in Figure 5. Three major polyphenols including chlorogenic acid, caffeic acid, and rutin are detected at the same retention time as the standards. The results of the content of three major polyphenols recorded at 275 nm are presented in Table 4. Chlorogenic acid was almost 10 times more abundant than caffeic acid in all extraction methods. Compared with separate enzyme extraction and ultrasonic extraction, ultrasound combined with enzyme extraction can significantly increase the content of rutin. In the meanwhile, the content of chlorogenic acid and caffeic acid obtained by UCE were both slightly lower than the other three methods. It can be seen that a certain degree of ultrasound does not affect the structure of phenols, and can also activate enzymes to release more active ingredients. In addition, the content of chlorogenic acid and caffeic acid extracted by SE was almost the same as that of UCE, but the extraction time is three times longer than that obtained by UCE, which required more energy. In addition, the content of the three qualitative HPLC standards is much lower than the TPC under the optimal process conditions compared with the Folin method. This may be due to the selected standards are mostly phenolic compounds. Other polyphenol compounds like flavonoids are not in the scope of this experiment. The content of quantified compounds in UCE was almost the same as González et al. [31], who reported that the content of chlorogenic acid was 28.45 mg/100 g, and the content of routine was 19.29 mg/100 g by Soxhlet extraction method, and they additionally detected the presence of quercitrin (10.40 mg/100 g) and epicatechin (8.69 mg/100 g). Other phenolic compounds like vitexin, isovitexin, isoorientinwere [1,32], and C-glycosylflavonoids [33] were also identified in passion fruit.

### 2.7. Infrared (IR) Spectroscopy

Infrared spectroscopy analysis is a powerful tool to characterize and identify functional groups present in compounds [34]. The infrared scanning spectrum of the purified extract is shown in Figure 6. The sample has strong absorption peaks near 3416 cm^−1^, 1643 cm^−1^, and 1122 cm^−1^. The absorption at the vicinity of 3400 cm^−1^ is relatively strong, wide and scattered, which is attributed to the stretching vibration of O-H, besides, the intermolecular hydrogen bonding shifts the peak to a low wavenumber [35]. The weak absorption at 2927 and 2855 cm^−1^ should be the C-H stretching vibration peaks of methyl and methylene groups. The band at 1643 cm^−1^ could due to stretching vibration of C=C of aromatic ring compounds [36], due to C=O stretching vibration of caffeic acid and its derivatives or other flavonoids [37], amino acids, and lipids [38]. The band at 1540 cm^−1^ indicated the characteristics of the benzene ring combined with the band at 1643 cm^−1^, showing the presence of phenolic hydroxy groups [39]. The bands at 1401 cm^−1^ and at 1122 cm^−1^ would be related to C-H bending vibration and to O-H bending vibration respectively [5]. It can be concluded that the diversity of functional groups indicate possible bioactive compounds like polyphenols and flavonoids are present in passion fruit extract, which further confirmed potential antioxidant activity [39]. However, the existence of qualitative polyphenols still requires HPLC to prove it.

## 3. Materials and Methods

### 3.1. Experimental Materials

The passion fruits were bought from an orchard in Guangxi, China. The passion fruits were dried in a vacuum freeze drier (TF-FD-1; Shanghai Toffon machinery equipment Co., Ltd., Shanghai, China) at −60 °C and powdered to particle (180 µm mesh size) in an electric blender (LFP-800; Yongkang Red Sun Electromechanical Co., Ltd., Shanghai, China). Then, the powdered sample was placed in a polyethylene bag.

### 3.2. Chemical and Reagents

DPPH (2,2-diphenyl-1-picryhydrazyl PubChem CID: 2735032) stable radical and ABTS (PubChem CID: 9570474); 2,2′-azobis (2-amidinopropane (ABAP, PubChem CID: 1969) were purchased from Sigma (St. Louis, MO, USA). Folin-Ciocalteu’s phenol reagent, sodium carbonate (PubChem CID: 10340), and cellulase were purchased from Macklin Biochemical (Shanghai, China). Deionized water was used. Eight standard phenolic acids, including gallic acid, chlorogenic acid, ferulic acid, caffeic acid, quercetin, pyruvic alcohol, rutin, and ellagic acid were purchased from Nature Standard (Shanghai Standard Technology Co., Ltd., Shanghai, China).

### 3.3. Extraction of Polyphenols from Passion Fruit

#### 3.3.1. Cellulase-Assisted Extraction

Cellulase-assisted extraction was carried out in an ultrasonic bath processor (KQ-500DB; Kunshan Ultrasonic Instrument Co., Ltd., Kunshan, China) at a constant frequency of 50 kHz. The equipment was furnished with a digital control system for ultrasound power, ultrasonic bath temperature, and time. The powdered sample (1.0 g) was dispersed in distilled water at a capped Erlenmeyer flask. After adding an amount of cellulase and adjusting the pH of the suspension, the reaction system was sonicated at different power, time, and temperature in the ultrasound apparatus with water. Then the crude extract was centrifuged, filtered, and stored at 4 °C. Each sample was measured in triplicate.

#### 3.3.2. Ultrasound Extraction

The powdered sample (1.0 g) was mixed with distilled water (50 mL) in a flask. The flask with suspension was sonicated at 30 °C for 47 min according to the optimization of cellulase-assisted extraction. Then the crude extract was centrifuged, filtered and stored at 4 °C.

#### 3.3.3. Soxhlet Extraction

According to the method shown by [40] with minor modifications. The extraction was carried out refluxing in a Soxhlet apparatus under the following experimental condition: the powdered sample (4 g) was mixed with distilled water (200 mL), extraction temperature (95 °C), ultrasound time (3 h). Then, the extract was obtained and stored at 4 °C.

#### 3.3.4. Cellulase Extraction

The powdered sample (1.0 g) was dispersed in distilled water (50 mL). The extraction condition was adjusted as follows: the pH value of 5, cellulase amount of 6%, and temperature of 30 °C. Then the crude extract was centrifuged, filtered and stored at 4 °C.

### 3.4. Determination of Total Polyphenols Content

The TPC of the passion fruit extract was determined according to the Folin-Ciocalteu assay reported by [5]. In this study, 1.0 mL of the diluted extract was mixed with 1.0 mL Folin-Ciocalteu’s reagent. Then, 1.5 mL 10% sodium carbonate was added in a tube and incubated at 25 ± 2 °C for 1 h. The absorbance was measured at 765 nm with an Alpha 1860 spectrometer (Lab-Spectrum Instruments Co., Shanghai, China). A calibration curve was built using a standard solution of gallic acid. The TPC was expressed as mg gallic acid equivalent (GAE)/1 g dried weight (DW).

### 3.5. Determination of Antioxidant Capacity

#### 3.5.1. DPPH Radical Scavenging Assay

2,2′-Diphenyl-1-picrylhydrazyl (DPPH) radical was used to evaluate the antioxidant activity of passion fruit extract, the method was described by [41]. Briefly, 4 mg of DPPH was dissolved in approximately 100 mL 95% ethanol and sonicated for 5 min to prepare a 0.1 mM/L DPPH solution, then the absorbance of the DPPH solution was adjusted to 1.200 ± 0.050 at 517 nm and 2 mL of DPPH solution was mixed with 0.2 mL of the sample and 0.8 mL of ethanol, gallic acid (20–100 μg/mL) was selected as a positive control substitute sample. The solution was incubated in dark conditions at room temperature for 30 min. The absorbance was measured at 517 nm.

#### 3.5.2. ABTS Radical Scavenging Assay

The ABTS assay was measured according to [42] with some modifications. Briefly, 10 mM phosphate-buffered saline (PBS) solution solved 12.5 mM of ABTS and 2,2-azobis (2-amidinopropane) dihydrochloride (ABAP), which pH was at 7.2. The mixture was completely blended in the incubator for 40 min at 68 °C and monitored under spectrophotometer, which the absorbance was 0.65 ± 0.020 at 734 nm. The scavenging activity was measured at 734 nm after 20 µL of sample and 980 µL of ABTS^+^ solution were mixed in a tube and then incubated for 10 min at 37 °C. 20 µL deionized water in 980 µL ABTS^+^ solution was considered as control consist.

By measuring the DPPH and ABTS^+^ scavenging activities of 20–100 mg vitamin C/L, the standard curves of the two assays were collected. The DPPH and ABTS^+^ scavenging activities were expressed as mg vitamin C equivalent (VCE)/100 g dried weight. All extracts were tested after being diluted 100 times to ensure the accuracy of the test results. Besides, each sample was done in triplicate.

#### 3.5.3. Single-Factor Experiments

The effect of extract was investigated by a single factor under the following parameters: liquid-to-solid ratio (10:1–50:1 mL/g), cellulase amount (1–5%), pH (3–7), temperature (30–70 °C), ultrasound power (200–500 W) and time (10–50 min). On the basis of the single-factor experiment results, four factors including liquid-to-solid ratio, ultrasonic time, ultrasonic power, and pH were selected as main variables for Response Surface Experiment. A four-factors-three-level experiment was designed by Box-Behnken principle in Design-Expert 8.0 (Stat-Ease, Inc., MN, USA) to optimize the passion fruit extraction process, and TPC, DPPH, and ABTS were used as response values. Table 5 shows the factors and levels.

#### 3.5.4. Response Surface Methodology Experiments

RSM was applied to optimize TPC and antioxidant activities from passion fruit. Based on the above single-factor results, three-level and four-factor were selected. The design matrix for a Box-Behnken design involved 29 design experiments (Table 1). Six replicated design points were used to evaluate the pure error. The response value (Y_i_) from BBD was showed using a second-order polynomial model given by the following equation:(4)$$Y_{\mathrm{TPC}}=\beta_{0\;}+\beta_{1}X_{1}+\beta_{2}X_{2}+\beta_{3}X_{3}+\beta_{4}X_{4}+\beta_{12}X_{1}X_{2}+\beta_{13}X_{1}X_{3}+\beta_{14}X_{1}X_{4}+\beta_{23}X_{2}X_{3}+\beta_{24}X_{2}X_{4}+\;\beta_{34}X_{3}X_{4}+\beta_{11}X_{1}^{2}+\beta_{22}X_{2}^{2}+\beta_{33}X_{3}^{2}+\beta_{44}X_{4}^{2},$$
where Y_i_ is the predicted value; *β*_0_ is a constant; *β*_1_, *β*_2_, *β*_3_, and *β*_4_ are the linear effect coefficients; *β*_12_, *β*_13_, *β*_14_, *β*_23_, *β*_24_, and *β*_34_ are the interaction effect coefficients; *β*_11_, *β*_22_, *β*_33_ and *β*_44_ are the quadratic effect coefficients; and X_1_, X_2_, X_3_, and X_4_ are the independent variables. Design-Expert software (version 8.0.6) was applied for the experimental design, data analysis and model obtained.

### 3.6. HPLC Analysis

The crude extract was diluted to 1 mg/mL aqueous solution and analyzed using an Agilent 1260 high-performance liquid chromatography (HPLC) system (Agilent Technologies, Santa Clara, CA, USA). This system includes degasser (G4225A), quaternary pump (G1311C), column temperature controller (G1316A), and DAD SL detector (G1314D). The chromatographic column was ZORBAX Eclipse Plus C18 (4.6 mm × 250 mm, 5 μm). The mobile phase included 0.1% phosphate acid in water (phase A) and acetonitrile (phase B). The flow rate of the mobile phase in the elution procedure was 1 mL/min. The column temperature was 30 °C. The injection volume was 10 μL. The samples were eluted according to the following gradient: 0–25 min, 10–45% phase B; 25–25.1 min, 45–10% phase B; 25.1–30 min, 10% phase B. Dual-wavelength detection was set at 271 and 320 nm. Samples for HPLC analysis were filtered through the 0.22 μm membrane filters. Eight standard phenolic acids, including gallic acid, chlorogenic acid, ferulic acid, caffeic acid, quercetin, pyruvic alcohol, rutin, and ellagic acid, were selected for qualitative and quantitative analysis.

### 3.7. Infrared Spectroscopy (IR)

The crude extract was purified and lyophilized into powder, then it was mixed with spectroscopic KBr (approximately 1:100), pressed into a sheet for IR analysis using FTIR spectrometer (Thermo Fisher Scientific, Waltham, USA). The FTIR spectrum was measured in the range of 4000 to 600 cm^−1^.

## 4. Conclusions

In the paper, the ultrasonic cellulase-assisted extraction method was successfully examined for natural polyphenols from passion fruit using a Box-Behnken design. In this work, four factors have been evaluated. The optimized condition was obtained, and fitted with the experimental value. ANOVA statistics suggested that the liquid-to-solid ratio is the most significant factor on TPC, DPPH, and ABTS antioxidant activity, and pH and temperature also play a significant role in DPPH and ABTS antioxidant activity. The predicted optimum conditions would obtain the maximum TPC, DPPH and ABTS antioxidant activity, respectively: 22.34 mg GAE/g DW, 415.37 mg VCE/100 g DW, and 465.42 mg VCE/100 g DW. The application of HPLC identified and quantified three major phenols including chlorogenic acid, caffeic acid, and rutin in the extracts. Compared three other extraction methods, UCE produced extracts that achieved better performance in the TPC and antioxidant activity tests in a shorter time. Based on the report, the ultrasonic cellulase-assisted extraction was an efficient method in the extraction of natural compounds from passion fruit.

## Figures and Tables

**Figure 1 molecules-26-02494-f001:**
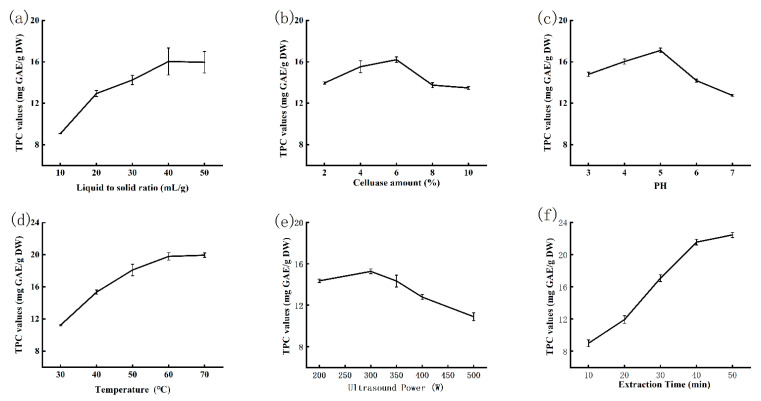
Effects of extraction parameters: (**a**) liquid-to-solid ratio, (**b**) cellulase amount, (**c**) pH, (**d**) temperature, (**e**) ultrasound power and (**f**) extraction time on the content of polyphenols from passion fruit. Data are presented as means ± SD, *n* = 3.

**Figure 2 molecules-26-02494-f002:**
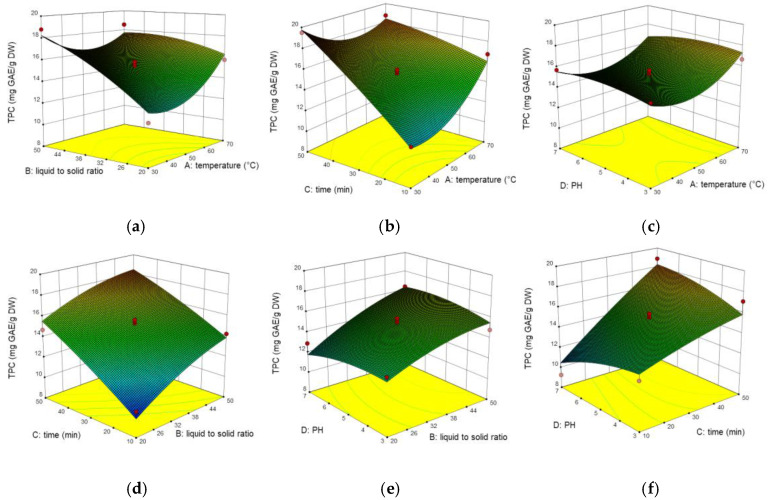
Three-dimensional response surface plots for the effect of (**a**) liquid to solid ratio/temperature, (**b**) time/temperature, (**c**) pH/temperature, (**d**) time/liquid to solid ratio, (**e**) pH/liquid to solid ratio, (**f**) pH/time on TPC.

**Figure 3 molecules-26-02494-f003:**
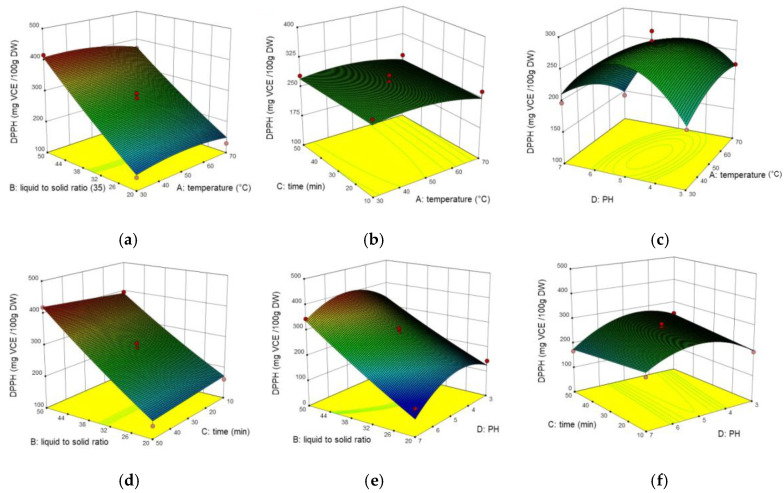
Three-dimensional response surface plots for the effect of (**a**) liquid to solid ratio/temperature, (**b**) time/temperature, (**c**) pH/temperature, (**d**) time/liquid to solid ratio, (**e**) pH/liquid to solid ratio, (**f**) pH/time on DPPH.

**Figure 4 molecules-26-02494-f004:**
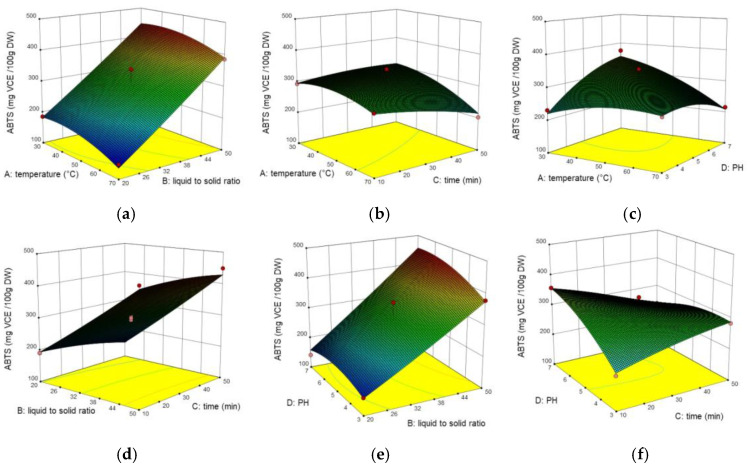
Three-dimensional response surface plots for the effect of (**a**) liquid to solid ratio/temperature, (**b**) time/temperature, (**c**) pH/temperature, (**d**) time/liquid to solid ratio, (**e**) pH/liquid to solid ratio, (**f**) pH/time on ABTS.

**Figure 5 molecules-26-02494-f005:**
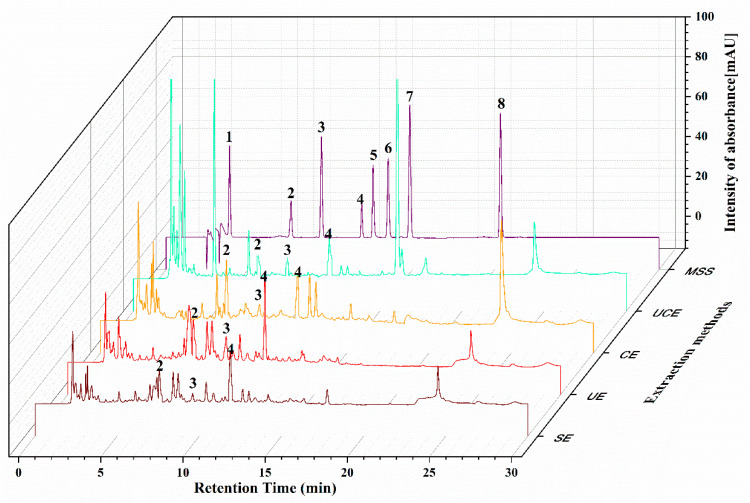
HPLC of passion fruit extracts obtained by CE, UE, UCE, and SE and mixed polyphenol standards. 1-Gallic acid, 2-Chlorogenic acid, 3-Ferulic acid, 4-Caffeic Acid, 5-Rutin, 6-Quercetin, 7-Ellagic acid, 8-Piceatannol.

**Figure 6 molecules-26-02494-f006:**
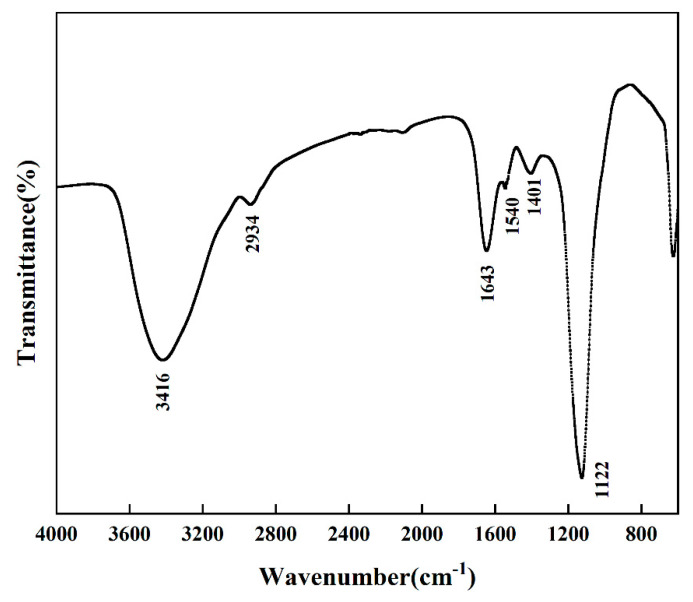
FTIR spectra of UCE extract.

**Table 1 molecules-26-02494-t001:** Response surface design and experimental results.

Run	X_1_ (Temperature, °C)	X_2_ (Liquid-to-Solid Ratio, mL/g)	X_3_ (Ultrasound Time, Min)	X_4_ (pH)	Y_1_ (TPC Value, mg GAE/g DW)	DPPH (mg VCE/100 g DW)	ABTS (mg VCE/100 g DW)
1	70	35	10	5	16.85	269.65	285.85
2	50	50	30	3	14.89	326.20	380.78
3	50	20	30	3	13.43	146.38	155.37
4	50	35	10	3	12.68	198.83	208.63
5	50	50	10	5	14.31	397.26	446.66
6	50	50	50	5	17.37	420.14	450.09
7	50	20	30	7	12.99	105.62	142.26
8	50	50	30	7	16.44	347.28	441.23
9	30	35	30	7	15.79	198.84	375.39
10	50	35	50	7	18.90	170.17	240.24
11	30	35	30	3	15.59	192.73	232.13
12	50	35	30	5	15.17	279.43	293.03
13	50	35	30	5	15.73	279.86	291.96
14	70	35	30	7	16.06	144.10	217.17
15	70	35	50	5	18.78	270.27	205.42
16	50	35	30	5	15.55	296.01	348.90
17	30	35	50	5	19.51	281.17	294.26
18	50	35	50	3	17.16	253.69	286.53
19	30	20	30	5	11.77	140.17	187.27
20	70	35	30	3	16.74	228.39	256.69
21	70	50	30	5	18.57	371.62	387.55
22	30	50	30	5	18.80	417.74	433.85
23	30	35	10	5	11.78	281.93	294.80
24	50	20	10	5	10.37	162.70	191.89
25	50	20	50	5	14.77	139.45	157.33
26	70	20	30	5	15.78	130.87	143.73
27	50	35	30	5	14.22	262.98	298.08
28	50	35	30	5	15.37	276.85	302.22
29	50	35	10	7	9.28	198.63	359.82

**Table 2 molecules-26-02494-t002:** The ANOVA for response surface quadratic models.

	TPC			DPPH			ABTS		
	Coefficient Estimate	F Value	*p* Value	Coefficient Estimate	F Value	*p* Value	Coefficient Estimate	F Value	*p* Value
intercept									
X_0_	15.21			279.03			306.84		
linear									
X_1_	0.80	8.01	0.0133 *	−8.14	2.33	0.1494	−26.77	24.65	0.0002 ***
X_2_	1.77	39.83	<0.0001 ***	121.25	516.20	<0.0001 ***	130.19	582.74	<0.0001 ***
X_3_	2.60	85.82	<0.0001 ***	2.16	0.16	0.6927	−12.82	5.65	0.0323 *
X_4_	−0.086	0.7644	−15.13	−15.13	8.04	0.0132 *	21.33	15.64	0.0014 **
Quadratic									
X_1_^2^	1.54	16.25	0.0012 **	−14.99	4.26	0.058	−21.44	8.55	0.0111 *
X_2_^2^	−0.53	1.9	0.1897	7.26	1.00	0.3343	4.04	0.30	0.5901
X_3_^2^	−0.24	0.4	0.5354	−0.44	3.56 × 10^−3^	0.9533	−8.02	1.20	0.2928
X_4_^2^	−0.47	1.51	0.2398	−67.07	85.36	<0.0001 ***	−23.68	10.42	0.0061 **
Crossproduct									
X_12_	−1.06	4.75	0.0469 *	−9.21	0.99	0.3362	−0.69	5.45 × 10^−3^	0.9422
X_13_	−1.45	8.89	0.0099 **	0.34	1.39 × 10^−3^	0.9708	−19.97	4.57	0.0506
X_14_	−0.22	0.2	0.658	−22.60	5.98	0.0283 *	−45.7	23.93	0.0002 ***
X_23_	−0.34	0.47	0.5023	11.53	1.55	0.2330	9.5	1.03	0.3265
X_24_	0.50	1.05	0.3238	15.46	2.80	0.1167	18.39	3.88	0.0691
X_34_	1.28	6.98	0.0193 *	−20.83	5.08	0.0408 *	−49.37	27.93	0.0001 ***
model		9.26	<0.001 ***		45.64	<0.001 ***		50.61	<0.001 ***
lack of fit		3.4	0.1245		3.07	0.1454		0.46	0.856
R^2^		0.928			0.979			0.981	

X_1_ = Temperature (°C), X_2_ = The ratio of solvent to solid, X_3_ = Ultrasound time(min), X_4_ = pH of solvent, TPC = Total polyphenols content (mg GAE/g DW), ABTS = 2,2-azobis (2-amidinopropane) dihydrochloride (mg VCE/100 g DW), DPPH = 2,2′-diphenyl-1-picrylhydrazyl radical(mg VCE/100 g DW), R^2^ = Coefficients of determination. Level of significance * *p* < 0.05, ** *p* < 0.01, *** *p* < 0.001.

**Table 3 molecules-26-02494-t003:** Comparison of the optimized UCE with other conventional extraction methods.

Extracting Method	Temperature (°C)	Liquid-to-Solid Ratio (mL/g)	Time (Min)	The Total Polyphenols Content (mg GAE/g DW)	DPPH (mg VCE/100 g DW)	ABTS (mg VCE/100 g DW)
UCE	30	50	47	21.03	420.50	453.83
UE	30	50	47	15.11	386.75	362.63
CE	30	50	47	10.08	250.85	225.86
SE	95	50	180	8.02	196.51	188.29

**Table 4 molecules-26-02494-t004:** Composition of passion fruit extracts obtained by optimized UCE and other three methods.

Polyphenol Compound	Retention Time	Regression Equation	R^2^	Extraction Method (μg/g; Dry Material)			
				UCE	UE	CE	SE
Chlorogenic acid	7.60	y = 27.208x − 53.724	0.9991	150.2	164.7	175.6	152.7
Caffeic Acid	9.46	y = 60.199x − 9.9358	0.9998	9.8	12.2	14.2	10.4
Rutin	11.95	y = 9.0995x + 6.5236	0.9997	187.5	1.4	49.1	68.1

**Table 5 molecules-26-02494-t005:** Independent variables and levels of the experiment.

Levels	Temperature	Liquid-to-Solid Ratio	Ultrasound Time	pH
−1	30	20	10	3
0	50	35	30	5
1	70	50	50	7

## Data Availability

The data presented in this study are available on request from the corresponding author.

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
