# Peer review of "Optimization of Ultrasonic Cellulase-Assisted Extraction and Antioxidant Activity of Natural Polyphenols from Passion Fruit"

_molecules, 2021, doi:10.3390/molecules26092494_

Round 1

Reviewer 1 Report

molecules-1173032-peer-review-v1

Optimization of Ultrasonic Cellulase-Assisted Extraction and

Antioxidant Activity of Natural Polyphenols from Passion fruit

The work should be reviewed in some of its sections and improved

Some suggestions are mentioned below

1- Figure 1 Foot note

Please revise the paragraph Data are presented as means+ SD

Change by means± SD

2- Section 4.7 HPLC Analysis

Eight standard phenolic acids, including gallic acid, chlorogenic acid, ferulic acid, caffeic acid, quercetin, pyruvic alcohol, rutin and ellagic acid, were selected for qualitative and quantitative analysis.

The company of origin of these patterns should be mentioned in this section or in the section4.2 Chemical and Reagents.

3- The reason for the selection of  eight standards should be mentioned in the section 2.5 HPLC Analysis.

4- Section Table 3. Comparison of the optimized UCE with other conventional extraction methods

The authors write “These results clearly demonstrate that UCE is a competitive and effective extraction technique for natural passion fruit properties, especially polyphenols.” Lines 206-208, page 8

In section Table 3 The total polyphenols content (mgGAE/gDW) the authors report 21.03 mgGAE/gDW

However, in the Table 4 Composition of passion fruit extracts obtained by optimized UCE and other three methods

The authors report a total content among the three major compounds of  .. approximately .350 µg/gDWA much lower value than the total phenolics determined by Folin-C methods

The authors should comment on what happened to the rest of the approximately phenolic compounds that they quantified by folin-C method

5- Section 2.6 Infrared (IR) Spectroscopy

Is to preliminary and lack of novelty, the paragraph should be revised , and improved.

6- Paragraph lines 174-178, page 6.

“As is shown in Figure 3c, the value of DPPH reached maximum as the temperature was 50 °C  and the value of pH was 5, but over this point, DPPH radical scavenging assay began to decrease. This may due to high temperature and pH may change the antioxidant  bio-compounds structure, thus influence the antioxidant ability.”

This should be supported by bibliographic references

7- In vitro antioxidant activity was estimated via DPPH and ABTS radical scavenging assays.

A paragraph in the results and discussion section referring to the results of the antioxidant tests and the quantified compounds should be included.

The work is weak in this section, a solid discussion with relevant bibliography is necessary.

Is there previous work published on the antioxidant potential and quantification of phenolic compounds?. They should be discussed.

After the suggested changes, the work may be considered for acceptance.

Author Response

We appreciate the time and effort that you dedicated to providing feedback on our manuscript and are grateful for the insightful comments on and valuable improvements to our paper.

Reviewer 2 Report

The authors of the manuscript "Optimization of Ultrasonic Cellulase-Assisted Extraction and Antioxidant Activity of Natural Polyphenols from Passion fruit" presented the data relating to a very thorough and timely study. I believe that some parts of the work should be better described because the reading does not fully understand the experimental design and how everything was carried out. Below is a series of indications and suggestions aimed at improving the manuscript and thus making it worthy of publication.

Introduction:

line 32: P. edulis must be in italics

line 36-37: I suggest to the authors of rewrites the sentence “The antioxidative passion fruit extract additionally posed in vitro tyrosinase inhibitory and SPF” because is not clear; in vitro must be in italics

line 54-56: the sentence “However, few studies about extracting polyphenols of passion fruit using ultrasonic cellulase extraction, especially research on how to extract polyphenols efficiently.” lacks the verb.

I state that I am not an expert in Box-Behnken design. Having said that, the experimental design relating to the data appearing in table 1 is not very clear to me: why none of the combinations of parameters is attributable to the combinations shown in figure 1? by choice, the parameters previously tested were not considered? Why does the temperature of 70 ° C appear in so many Runs, when it was previously stated that it is better not to exceed 60 ° C?

What are the values obtained from the combinations of figure 1 used for?

I believe that the IR analysis relating to a multi-component extract is almost completely superfluous, especially when the main constituent components of a certain extract are known. HPLC is certainly a more powerful tool, possibly coupled to a mass, the use of which is aimed at identifying compounds. Knowing which functional groups are present in a well-known plant extract is of little value, but that's just my opinion.

Line 285: why were the samples incubated at 25 ° C and not at the optimum temperature of 37 ° C typical of the Folin test?

DPPH assay: at what concentration were the samples tested? the results shown refer to identical concentrations of the various samples?

Since the DPPH test is highly dependent on concentration, if one wishes to compare results expressed in mg VCE / 100g DW and not as IC50, it is essential that all samples are tested at the same concentration. Either the concentration relative to the result is indicated, or alternatively it can be stated that all the values obtained are, for example. related to a radical inhibition of 50%.

It is the sentences of the authors "The absorbance of the solution was adjusted to 1.200 ± 0.050 at 517 nm" and "0.2 mL of each sample extract" that made me doubt.

The unit of measurement of the test and the standard used are not indicated.

Author Response

(The authors gave the same response as above.)

Reviewer 3 Report

Minor remarks

The references should be prepared according to the Instruction for Authors.

In the y-axis of Figure 1, the unit is not proper. It should be presented as follows: mg GAE/g DW.

The resolution of Figures 2, 3, and 4 should be improved.

All other minor remarks are given in the manuscript.

Major remarks

In introduction section, the optimization of extraction process using the experimental design should be discussed. It is necessary to explain the advantages of the experimental design approach compared to other optimization techniques. The following references are relevant in this scientific topic so that they can be included in this section: DOI: 10.1007/s11081-020-09565-0; DOI: 10.3390/biom11020225; DOI: 10.1007/s13197-020-04312-w.

The obtained results should be better discussed and compared with available data. In this way, the manuscript is presented as a laboratory report.

Author Response

(The authors gave the same response as above.)

Round 2

Reviewer 1 Report

The authors have responded satisfactorily to the suggestions made. The manuscript should now be accepted for publication in its current state.